# Parents’ Perceptions about Salt Consumption in Urban Areas of Peru: Formative Research for a Social Marketing Strategy

**DOI:** 10.3390/nu12010176

**Published:** 2020-01-08

**Authors:** Vilarmina Ponce-Lucero, Lorena Saavedra-Garcia, Erik Cateriano-Arévalo, Silvana Perez-Leon, David Villarreal-Zegarra, Diego Horna-Alva, J. Jaime Miranda

**Affiliations:** 1CRONICAS Center of Excellence in Chronic Diseases, Universidad Peruana Cayetano Heredia, Lima 15074, Peru; 2School of Medicine, Universidad Peruana Cayetano Heredia, Lima 15102, Peru

**Keywords:** perceptions, social marketing, salt, hypertension and Peru

## Abstract

Background: Salt intakes in Latin America currently double the World Health Organization’s recommendation of 5 g/day. Various strategies to reduce the population’s salt consumption, such as raising awareness using social marketing, have been recommended. This study identified parents’ perceptions of salt consumption to inform a social marketing strategy focused on urban areas in Peru. Methods: Using a sequential exploratory methods design, parents of pre-school children, of high and low socioeconomic status, provided qualitative data in the form of interviews and focus groups. Following this, quantitative data was obtained via questionnaires, which were sent to all parents. The information was analyzed jointly. Results: 296 people (mean age 35.4, 82% women) participated, 64 in the qualitative and 232 in the quantitative phase of the study. Qualitative data from the first phase revealed that the majority of mothers were in charge of cooking, and female participants expressed that cooking was “their duty” as housewives. The qualitative phase also revealed that despite the majority of the participants considered their salt intake as adequate, half of them mentioned that they have tried to reduce salt consumption, and the change in the flavor of the food was stated as the most difficult challenge to continue with such practice. Quantitative data showed that 67% of participants would be willing to reduce their salt intake, and 79.7% recognized that high salt intake causes hypertension. In total, 84% of participants reaffirmed that mothers were in charge of cooking. There were no salient differences in terms of responses provided by participants from high versus low socioeconomic groups. Conclusions: The results point towards the identification of women as a potential target-audience of a social marketing strategy to promote reductions in salt intake in their families and, therefore, a gender-responsive social marketing intervention is recommended.

## 1. Introduction

Globally, hypertension prevalence is increasing. In low-and-middle-income countries, the burden of this disease is caused by nutritional transition and inadequate healthcare. Hypertension is the main cause of cardiovascular disease, the leading cause of death and disability worldwide [1]. In Latin America, hypertension prevalence is 44.0%, with the highest rates in Brazil (52.5%) [2].

High salt intake is associated with hypertension, a chronic condition that causes millions of early deaths in the world [3,4]. Studies showed that 10% reduction of salt at the population level can reduce hypertension prevalence, and thereby reduce cardiovascular risk [5]. In many Latin American countries, average population salt intake is double that of the World Health Organization’s (WHO) recommended maximum daily intake for salt, 5 g/day [6]. There is limited information about the salt consumption in Peru at the national level, one study in the North of Peru reported salt consumption levels of 11 g/day (4.4 g of sodium) [7].

The taste for salt is established through diet at a young age. Parents and caregivers can help lower sodium by influencing the way foods are produced, sold, prepared, and served [8]. Biologically, young children are more prone to choose and eat salty flavors, but this preference can be modeled by social factors, such as their parents feeding practices [9,10]. To promote salt reduction and consequently promote a healthy habit, parents of preschool-aged children were selected because early childhood is a lifespan stage where children learn about food and eating, while transitioning to an omnivore diet [11].

The World Health Organization (WHO) states that the strategies to reduce salt intake comprise of government policies, promotion of healthy foods in different communities, and increased public awareness through social marketing [12]. Different organizations have presented evidence that social and community interventions can reduce salt consumption [13,14]. Social marketing is a strategy for behavioral change characterized by the use of concepts, principles, and marketing techniques, through a systematic planning process [15,16].

Social marketing is gaining traction towards becoming a standard approach to public health challenges [17], having effective results in changing consumption habits in the populations to which it has been applied [18,19]. There is particular evidence for the reduction of salt consumption in Brazil and England [20,21,22].

One of the core components in the design of a social marketing intervention is the exploration of the insights that drive behavioral changes within the audience of interest [23]. In Peru, determinants of dietary habits, food choices and its use, similar to other contexts, are influenced by cultural factors and traditions [24,25]. For this reason, the study of people’s understanding of salt intake and health outcomes is crucial to identify the triggers of behavior change at individual and community level and ensures better design and implementation of appropriate interventions.

In this study, qualitative and quantitative methods were employed to gain an in-depth understanding of the knowledge level, attitudes, current behaviors, and practices associated with salt consumption. The barriers and motivators that influence behavioral changes to promote salt reduction in food preparation among parents in Peru are additionally explored in this study. Potential differences between different socioeconomic groups were also assessed. In doing so, the findings from this study will inform the development of a social marketing strategy targeted towards urban areas in Peru.

## 2. Materials and Methods

### 2.1. Study Design

The present study had an exploratory sequential mixed-methods design. In the first phase, a qualitative phase, in-depth interviews and focus groups were used. Then, in the second phase, a self-administered survey was designed to obtain quantitative information from a wider group of participants. Subsequently, both types of data were jointly analyzed.

### 2.2. Setting

This study was conducted in three districts that were either of a high or low socioeconomic status (SES) in Peru: Miraflores and San Miguel in Lima (high socioeconomic status) and Mi Peru in Callao (low socioeconomic status) [26]. In order to show possible differences in dietary patterns or practices, in each district, private and public kindergartens with parents of preschool-aged children were selected to participate in the study.

### 2.3. Study Population

The aim was to collect information from the urban population differentiated by socioeconomic status; participants of study were parents (men and women) of preschool-aged children.

According to official estimates of the Miraflores populations [27], in 2017 there were 99, 337 inhabitants with a life expectancy of 76 years, women illiteracy rate 0.2%, and the incidence of monetary poverty is 1.5%. In the Mi Peru district [28], there were 59,005 inhabitants, with a life expectancy of 78 years, a women illiteracy rate of 3.2%, and the incidence of monetary poverty in a range between 23.2% and 29.2%.

The authors had a hard time finding men in participating in the focus group in Miraflores and, as a result, a focus group of men was conducted in San Miguel, a district with similar characteristics to Miraflores.

Sampling was non-probabilistic, given that the nature of the study was not focused on ensuring representativeness of the sample but on the exploration of particular characteristics around salt consumption.

### 2.4. Procedures

#### 2.4.1. Recruitment

In the first phase, participants were recruited through an invitation made by the principals or teachers of the kindergartens. Interviews and focus groups (four–eight parents each) were carried out in classrooms. In each focus group, one researcher moderated, while another observed and took notes. Additionally, participants filled out a short questionnaire to gather sociodemographic information.

For the second phase, a self-administered questionnaire was sent through a lesson book and only one of the parents was asked to complete it. The questionnaire was anonymous.

#### 2.4.2. Data Collection

Data collection was carried out between December 2017 and January 2018. All of the data collection tools were designed and validated by a multidisciplinary research team in a series of meeting discussions. The toolkit included an in-depth interview guide, focus group guides (semi-structured and open-ended questions), and a survey. The interview and focus group guides covered the following topics: (1) decision-making of meals at home, (2) food preparation practices at home, (3) views of salt consumption, and (4) relationships between salt intake and health. The interview guide included one more section to collect the opinions about potential promotional materials (See Appendix A). The focus group guide was piloted in two focus groups of women and men, developed in a public kindergarten located in a low-status area in the district of Chorrillos.

A self-administered questionnaire was designed using the information that was collected at the semi-structured interviews and focus groups. The goal was to validate the information participants reported in the first phase. The questionnaire was divided into five sections: (1) sociodemographic characteristics; (2) salt consumption in their family; (3) meals preparation and planning; (4) relation between salt and health; (5) spokespersons, content of promotional materials, and channels to promotion messages. (See Appendix A).

### 2.5. Data Analysis

In-depth interviews and focus groups were recorded with the permission of participants, and were transcribed and analyzed using qualitative analysis software ATLAS-ti 8 (Scientific Software Development GmBH, Berlin, Germany). A preliminary list of codes and definitions by categories was developed a priori according to the questions of the instruments. Transcripts of one focus group and one in-depth interview were coded by the research team (five people). Then, some transcripts were assigned randomly to each researcher who checked out the coding using the final list of codes. These sets of codified transcripts were checked by another researcher of the team in two sequential rounds of revision, one for focus groups and one for interviews. Afterwards, data were summarized, taking into consideration gender, district, and type of school.

Data obtained from the self-administered questionnaire was analyzed descriptively and with the chi-squared test for evaluating the association between variables.

### 2.6. Ethics

Ethical approval was obtained from the Institutional Review Board of Universidad Peruana Cayetano Heredia, Lima, Peru (project code 66215). Written informed consent was obtained from all participants before interviews and focus groups took place. The questionnaire was anonymous.

## 3. Results

### 3.1. Phases and Number of Participants

In the qualitative phase, seven focus groups were conducted: one in San Miguel, four in Mi Peru, and two in Miraflores, with an aggregated total of 40 participants (Table 1). Focus groups were conducted separately by gender and type of preschool (private or public).

Additionally, 24 in-depth interviews with women were conducted, 14 in private and 10 in public kindergartens for each district, and four with men in San Miguel.

For the quantitative phase, 100 self-administered questionnaires were delivered to each preschool in three districts; 400 questionnaires were expected to be received but only 238 questionnaires were returned (Table 1). Six participants had missing values in the sex variable and were excluded, only 232 participants were then considered for the analysis, 59.5% was the participation rate.

### 3.2. Participant’s Characteristics

A total of 296 participants took part in the study, of which 82% women, with an age range 18 to 61 years (mean age 35.4). The majority was married (40.7%), employed (79.1%), and had a university or technical qualification (35.1%). The sociodemographic characteristics are presented in Table 2.

### 3.3. Cooking Practices

#### 3.3.1. Qualitative Phase

In most cases, participants reported that female family members were mainly responsible for home cooking. There were a few cases, primarily in the districts of high socioeconomic status, where men were also more frequently found to share or be in charge of this task. Additionally, in most cases, the person in charge of cooking was the one who decided what was going to be cooked.

Half reported being the sole decision makers for family meal planning, and a third pointed out they would ask family members for input on what they wished to be prepared. Only a small number of participants mentioned that choosing what was going to be cooked completely depended on the decisions made by their husbands or children: “I can ask my husband what he wants me to prepare for you, but the one who makes the decision is me” (Interview 18, Woman, Miraflores).

Furthermore, in seven of the focus group discussions with women, when asked “who was in charge of buying the groceries?” they mentioned that they were the ones in charge of buying the groceries. In one focus group, it was mentioned that cooking and buying their groceries was their “duty” as housewives (“mi deber”). “That is, we drop our children [at school], we go to the market, and return home to cook. (...) That is our duty every day (laughs).” (Focus group 2, Women, Mi Peru).

In the few cases that reported either being solely or jointly responsible for cooking or where it was a shared task, it appeared that the task of cooking went along with the decision of what was going to be cooked and buying the groceries.

Additionally, more than half of the participants said they would usually eat home-cooked meals, and this included taking food cooked at home to work. In a few cases, some members of the families, especially the father, would eat outside of their homes in local restaurants with set lunch menus near their work.

There were no clear differences in the responses provided by participants from different districts and different socioeconomic groups. But when asked how often the family would eat out, differences were evident between families of high socioeconomic status, with more than half of them reporting eating out every weekend. This was in contrast to families from districts of low socioeconomic status that would sporadically go out to eat, in most cases, only for special occasions. Furthermore, comparing the families with children in public or private kindergartens, the latter would report eating outside their homes more frequently.

In the interviews and focus groups, parents were asked what was the most important criterion while cooking, the options given to the participants were “health”, “taste”, “time”, or “money”. Most of the informants explained they prioritized having a healthy meal over the other characteristics. Some interviewees explained that having a healthy meal went along with making it tasty because if their children did not like the food they would not eat it. “In my case, [I choose] the taste and healthy. Because as healthy as the food can be, my son will not eat it without taste.” (Focus group 3, Women, Miraflores).

#### 3.3.2. Quantitative Phase

The self-administered questionnaires found that female family members were primarily responsible for cooking, with 65.5% of men and 86.7% of women indicating that mothers are the ones who decide what to cook. This was followed by as stated by 13.8% of men and 5.3% of women; finally, only a small group said that it was the parents who made the decision on what to cook (3.4% of men and 1.6% of women).

Those who decide what will be prepared reported that they receive suggestions mainly from their partner (56.3% of men and 32.8% of women) or their children (9.4% of men and 20.1% of women), while 40.2% of women and 28.1% of men said they did not consult other people when deciding what to cook.

The most important criteria reported by the participants to decide which meals to cook were that the meal is healthy (62.1% in women and 46.7% in men) and that the meal has a short preparation time (15.8% in women and 40% in men). The least valued criteria to decide what to cook was the money required for its preparation (13.2% in women and 6.7% in men) and taste of food (8.9% in women and 6.7% in men).

### 3.4. Perception of their Consumption of Salt and Seasonings

#### 3.4.1. Qualitative Phase

Most of the participants from interviews and focus groups did not consider that their consumption of salt was high; most would even consider their consumption of salt to be low. “Well, I really eat very little salt. I never cook food that is too salty; on the contrary, normally it lacks salt (laughs).” (Interview 36, Woman). Some people would say they recognized their food was low on salt because they would receive comments from families or friends (see Appendix A).
“Interviewer: And someone who has visited you has said “it is low on salt”?Woman: My friends, sometimes my friends.Interviewer: What have they told you?Woman: Eh, can you pass me the salt [saltshaker]? It [the food] lacks salt. And I pass them the salt; I say here, use this [saltshaker].” (Interview 16, Woman).

In other cases, they would express their consumption of salt was low because they felt other people’s meals were too salty. On the other hand, only two people who were interviewed (one man and one woman) said they thought their consumption of salt was high: “In my house, my husband consumes a lot of salt. I think the problem is me because sometimes I use high [amounts of] salt and I don’t taste it, because I am used to it. I think that I use too much salt when cooking.”(Interview 18, Woman).

Of the total of participants that said they had low or moderate salt consumption, almost all said they used artificial flavorings, the most common seasoning being “Ajinomoto” (a popular brand of monosodium glutamate in Peru), with more than half of the total number of people interviewed reporting using it. Some of the participants that did not use artificial flavorings said these were “chemicals” and could be harmful to their health; most of them were women from a high socioeconomic status district.

The interviewees also used natural seasonings and herbs to provide flavor to their meals. During the interview and focus groups, some images were shown to the participants in order to know their familiarity with the ingredients. The most commonly known ingredients to participants, (where at least 50% of the interviews recognized them), were ginger, mushrooms, garlic, and herbs. Despite being familiar with them however, participants were only able to point out a limited number of dishes where they could be used.

Moreover, over half of the participants, both from the interviews and focus groups, stated that they had attempted to lower their consumption of salt. While some reported being successful in achieving this, a number of participants said they had struggled or had not managed to reduce their salt consumption. Several participants had attributed their attempt to lower their consumption of salt to health-related issues, for example, hypertension, gallstones, diabetes, pancreatitis, or constipation.

Others would mention that they done so because they had wanted to lose weight or because they thought consuming too much salt was “bad” or dangerous for them and their families. Of the participants that had never tried to lower their salt consumption, some mentioned that they did not believe it was necessary as they considered their current consumption of salt appropriate or low.

In addition, the majority of the people that had previously tried to lower their salt consumption had reported that they had attempted to do so by replacing table salt for soy sauce, another high sodium product, but had reported having difficulties during this process due to the drastic change in taste.
“I started to do it, I got used to adding less salt to the food. Normally it is not necessary to add salt to the food because we use Sillao (soy sauce), and Sillao has a high salt content, so it also gives flavor meals. But it did cost us at first because we felt like it was unpleasant, but then we got used to it.” (Interview 12, Woman).

Moreover, some would say the main difficulty was the rejection of their family members, stating that sometimes they would add more salt to their meals after being served: “My father, he does not complain, but he always adds more salt to his meals. He always does it, whether he’s eating at home or eating out, and to whatever food he eats.” (Interview 13, Woman).

The strategies participants used to lower the consumption of salt were diverse, some would say they reduced the addition of salt gradually, would not put salt in the rice or change from tablespoon to teaspoon to measure the amount of salt, and in certain cases, they had changed their consumption of salt by using alternative ingredients (soy sauce or margarine): “When I was younger, to avoid adding more salt, I cooked the rice or pasta with margarine and, then it was finished off with a pinch salt.” (Interview 12, Woman).

#### 3.4.2. Quantitative Phase

With regards to their perception of salt intake, 23.3% of women and 10.3% of men who answered the questionnaire believed that their consumption was low.

In terms of the products used for seasoning, 50% of men and 28.6% of women reported that they used artificial condiments; 30.3% of men and 22.4% of women said they used daily to cook, and 33.2% of women and 33.3% of men responded that they rarely used them when cooking. In contrast, 46.7% of men and 57.1% of women said they use natural condiments for cooking.

### 3.5. Knowledge about the Consequences of High Salt Consumption

#### 3.5.1. Qualitative Phase

Awareness of the risks of excessive salt consumption was high, with only two people not knowing that salt had an impact on their health. The three most common reasons of why salt could be harmful were because it could have an impact on their blood pressure, it could have an impact on the kidneys because they “retained fluids”, and some suggested it could lead to weight gain. Other reasons mentioned with less frequency were: it could have an impact on the heart, cause swelling (“hinchazón”), and it could have an impact on the liver or cholesterol.

Few people knew whether there were any positive aspects of consuming salt, with most saying salt was useful because it gave flavor meals. Some participants did say that salt was good because it could help the wounds to heal, because it had iodine or because the body also needed salt.
“Because if you stop consuming salt, your body is going to need it, right? Anyways, we have to consume at least some sodium, haven’t we? But... excessive salt intake is bad, isn’t it? Then, we need to know how to balance it: not too much or too little salt, right? Your body needs salt. I have read about that somewhere before.” (Interview 26, Woman).

When asked if participants knew what the relationship was between salt and hypertension, almost a third did not know, one third knew there was an association, but could not explain what it was, and around one third knew salt was related to high blood pressure. Some of the participants that identified the salt-hypertension association did so because they had a relative that had the disease and they had been told to consume less salt as a result.
“Look, I have heard about it because my grandmother suffered from [hypertension], she was a patient with hypertension, and elderly members in my family also have hypertension. So my family has become used to eating food that is normally low in salt. Always.” (Interview 11, Woman).

#### 3.5.2. Quantitative Phase

The self-administered questionnaire found that approximately 79% of men and women said that hypertension was the most commonly linked illness to a high salt intake, and 9.6% of men and 10% of women did not know what disease is related to an excessive salt intake.

### 3.6. Motivators and Barriers to Reduce Salt Intake

#### 3.6.1. Qualitative Phase

The methods the participants were willing to use to lower their salt consumption was also explored. In general, focus group participants and interviewees were interested in reducing their salt intake and the majority of them explained how they would do it, except two people, who were not able to reduce their salt intake. Some participants mentioned they would adopt the new dietary habit where they would gradually reduce the amount of salt they usually use, while others would use alternative ingredients to flavor food instead of using salt when cooking. In both cases, the two main motivators for adopting this new behavior were the influence of family members and having a disease that affected them. Some of the barriers that were found to reducing salt intake included the influence of family members and changes in food taste.

#### 3.6.2. Quantitative Phase

The questionnaire asked if people were willing to reduce their salt intake without altering the taste of their food; 79.3% of men and 65.1% of women reported that they would agree to reduce their salt intake on this condition, followed by 10.3% of men and 11.1% of women who answered “maybe”.

When asked what they would feel (fear, concern, guilt, and other) if they knew their children could become sick with a chronic disease such as hypertension from consuming too much salt, 77.4% of men and 82.5% of women responded “concern”.

Another key fact was that 21.6% of questionnaire participants perceived their consumption of salt to be low.

## 4. Discussion

The goal of this study was to explore knowledge, attitudes, and behaviors around salt consumption from foods prepared at home. In addition, this study aimed to explore barriers and motivators that influence behavior change in order to inform the design of a social marketing strategy tailored to promote the reduction of salt consumption.

### 4.1. Main Findings

Our study shows that parents are interested in the health and welfare of their families; however, changes in salt consumption represent more than a health issue, as they are also linked to taste and food preferences. Additionally, most of the participants were aware that high salt consumption has implications for their health, but only a third reported knowing the direct relationship between high salt consumption and hypertension.

Across all socioeconomic groups, mothers are generally the ones that decide what their families eat at home, even when they are not the ones who prepare the food. This confirms the important role mothers have when considering the diet of their families, especially among families that do not eat frequently outside their homes.

For mothers, the acceptance of the food they prepare by other family members is crucial, and this is closely linked to taste. In fact, mothers mentioned their concern that regardless of how healthy a meal is, if it is not tasty, their kids will not eat it. Additionally, when they try to reduce salt in homemade food, for example in cases motivated by a family member having a disease, the main difficulty to engage and maintain this behavior was taste. Therefore, taste appears as the principal barrier to reducing salt intake. Similar results are found in the formative research that lead to the “Skip the Salt, Help the Heart” campaign, where authors reported that people associate a low-sodium diet with bland or tasteless foods [29].

Moreover, the results show that most mothers usually add artificial flavorings. Considering that those artificial flavorings are high in sodium, and based on the findings of a recent study that suggest the use of artificial seasonings as a proxy of salt intake [30], our results point towards a high consumption of sodium among these families. These findings should be a concern for government and health authorities, given that most of the participants believe they consume a moderate amount of salt. Our findings are supported by other studies in developing countries, where less than 15% of participants consider they consume too much salt [31,32].

Natural seasonings and herbs were recognized by most of the participants; however, participants only mentioned a few number of dishes where they would be used. The high use of artificial seasonings in the diet, and low use of the natural ones, could be explained by the fact that time is an important determinant of what people decide to cook or eat. In the qualitative phase of this study, some people mentioned time as a criterion to decide what to eat; in the quantitative phase, this was the second most common answer after healthiness.

In addition, 81% of the participants indicated that they would feel concerned if their children became unwell due to excessive salt consumption, and this observation could be one of the reasons explaining why participants considered health as the most important criterion when deciding what foods will be prepared or cooked at home. Similar and related concerns, e.g., inexperience in planning and meal preparation and time commitment, have been reported as the most common barriers to make healthy choices for their families [33].

### 4.2. Implications for a Social Marketing Campaign

The results of this study yield important insights for public health practitioners and social marketers interested in promoting a sustained decrease in the level of salt consumption in Peru. As the results show, mothers are not only responsible for cooking or buying groceries, they also decide what will be prepared. Therefore, mothers are an important audience for a social marketing campaign focused on reducing salt intake in homemade food. Previous studies have reported effective changes in the diets of families that have implemented similar interventions when mothers were selected as a primary audience [34].

On the basis of the social marketing principles [35], we believe that it is important to critically evaluate the implications of a potential strategy that solely focuses on mothers, particularly on the unintended reinforcement of gender stereotypes [36,37], while taking advantage of the key role that women play in potentially influencing larger family groups. This reflection arises from two key observations: first, among mothers, there is a belief that it is their “duty” to do household chores, and second, participating men said they shared the task of buying the food with their wives. This mismatched observation in our findings may suggest that men consider themselves as having a more active role in deciding the meals and in the buying of groceries, in comparison to what women would express in their testimonies.

The main barrier to the reduction of salt consumption was linked to the “lack of” taste in food that was low in sodium, a common feature across both socio-economic status and gender. Additionally, people’s lack of knowledge on the alternatives to salt and the negative effects they have on health e.g., soy sauce, stock cubes etc., are all high in sodium, which is ultimately bad for health.

Therefore, a social marketing campaign should approach the problem by promoting ways to cook with less salt while still retaining flavor, for instance using natural seasonings. As such, this would imply having two related outcomes; first, to reduce the salt in homemade food, and second, to promote the use of natural seasonings. In this sense, the desired behavior should be promoted within a set of behaviors associated with reducing salt intake, and not just using less salt when cooking.

While reducing salt intake at home can be challenging [38], participants in our study mentioned several ways to reduce salt intake when cooking, such as using a small-size spoon to add salt or replacing salt with other ingredients. This suggests that parents show a good predisposition to adopt a new technique when cooking to reduce their salt intake. The insights of leveraging parents’ desire to cook with creativity can also be part of a social marketing strategy.

Additionally, the campaign should avoid focusing on the health benefits of low salt consumption given that the association between high salt consumption and certain diseases does not resonate with most participants. Even when participants reported that they have tried to reduce salt due to a disease of a relative, they have reported difficulty in eating healthily due to the change of taste. However, it is also important to consider including an educational campaign not only to raise awareness about the health consequences of high salt consumption but also to generate synergies and increase the effectiveness of the interventions.

Natural seasonings and herbs to enhance taste are alternatives well known by participants, but are less used, probably due to their lack of knowledge of different recipes where they can be utilized. These ingredients have the benefit of being healthier and contribute to the prevention of non-communicable chronic diseases [39]. For example, a cookbook with a variety of recipes that includes accessible natural seasonings and herbs could be included in the campaign to promote its use.

### 4.3. Strengths and Limitations

The main strength of this study was the two-phase mixed methods approach, where the information collected through a survey was used to confirm and contrast the data collected from the qualitative methods. Another strength was securing the participation of parents from two different socioeconomic groups as well as public and private kindergartens to capture the diversity of perception and practices among different informants.

The study also has limitations. Firstly, the actual amount of salt intake was not measured quantitatively; thus, it cannot be compared with the answers provided by participants about their salt consumption. Secondly, our study protocol did not consider approaching parents for in-depth interviews to better capture their behaviors and insights related to decisions regarding food. This decision was made a priori on the assumption that fathers would not be available. This was later confirmed when conducting the focus groups, when there were fewer male participants than women.

Moreover, all the methods could be biased by the self-reported nature of the information retrieved, and other qualitative tools such as journey mapping could complement the information generated. However, the combination of insights provided by focus groups, interviews, and questionnaires provide a strong enough approach to gain an adequate understanding of the actual behaviors around salt consumption.

### 4.4. Conclusions

Salt and its influence in the taste of food are determinants of dietary patterns and food choices. Therefore, to achieve a reduction in salt consumption, it is not enough to carry out dietary recommendations or educational campaigns alone. It is necessary to know and understand the motivators and barriers that are not always considered when promoting new behaviors, as well as interests, practices and people that can drive and make the reduction of people’s salt consumption sustainable.

This study showed that women play a key role in their family’s diet. The main barrier for the adoption of a desired behavior was the change in taste of meals prepared at home. By contrast, there was not a consistent response for motivators in reducing salt intake and maintaining the change. A social marketing strategy should select mothers as a target audience. Furthermore, the strategy ought to promote salt reduction without sacrificing taste through the use of natural alternatives.

## Figures and Tables

**Table 1 nutrients-12-00176-t001:** Phases and number of the participants.

District	Kindergarten	Sex	Focus Group (n = 40)	In-Depth Interviews (n = 24)	Questionnaire (n = 232)
Miraflores	Public	Women	8	5	70
		Men	7	−	13
	Private	Women	−	5	26
		Men	−	−	5
San Miguel	Private	Women	6	−	30
		Men	−	4	6
Mi Perú	Public	Women	6	5	54
		Men	4	−	6
	Private	Women	5	5	18
		Men	4	−	4

**Table 2 nutrients-12-00176-t002:** Sociodemographic characteristics of the participants.

		Men (n = 53)	Women (n = 243)	Total (n = 296)
District	Miraflores Private	9.4%	12.8%	11.9%
	Miraflores Public	7.7%	34.1%	34.8%
	San Miguel Private	18.8%	14.8%	15.6%
	Mi Peru Private	15.1%	11.6%	12.2%
	Mi Peru Public	18.8%	26.7%	25.5%
Civil Status	Single	16.9%	12.4%	13.6%
	Married	34.0%	42.8%	40.7%
	Cohabitant	41.5%	35.0%	36.4%
	Separated	3.8%	7.8%	7.0%
	Divorced	3.8%	1.2%	1.7%
	Widower	0.0%	0.4%	0.3%
	Missing	0.0%	0.4%	0.3%
Education Level	No level	0.0%	3.7%	3.0%
	Primary	22.7%	30.5%	29.8%
	High school	32.1%	30.9%	30.8%
	University	35.8%	25.9%	27.5%
	Institute	7.5%	7.8%	7.6%
	Missing	1.9%	1.2%	1.3%
Currently Working	Yes	86.8%	77.4%	79.1%
	No	11.3%	21.4%	19.6%
	Missing	1.9%	1.2%	1.3%

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
