# Peer review of "Parents’ Perceptions about Salt Consumption in Urban Areas of Peru: Formative Research for a Social Marketing Strategy"

_nutrients, 2020, doi:10.3390/nu12010176_

Round 1
Reviewer 1 Report
Summary
This paper aimed to determine parent perceptions around salt in order to design a social marketing strategy to lower salt intake. High salt intake is a major public health issue around the world and strong, national level actions are required. Novel interventions, such as social marketing studies, have been highlighted as potentially useful in lowering salt intake. This paper highlights an important step in the design of such an intervention.
The article is much clearer and well-organised in its revised form, however there are still some issues that require further revision. Some text within the Materials and Methods and the Discussion belongs in the Introduction, in order to give appropriate context and justification for the study. Justification for the study population is still lacking and must be addressed. The study would be much stronger if it involved a representative sample of the population and so the present study population must be justified more.
The results are much clearer and well laid out and the discussion provides a convincing argument for the need for a social marketing strategy.
Specific Comments (PDF notation attached)
Line 24 – What is the 84% in reference to? Is it from the quantitative phase?
Line 39 – If Peru has the lowest rates of hypertension in the region, why is it necessary to lower salt intake in the country? More justification required here
Line 39/40 – this sentence could be moved to Line 43 (prior to the sentence starting ‘In many Latin American countries…’)
Line 42 – Definition of salt not required
Line 44 – ‘average population salt intake’ reads better than ‘the average salt intake in populations’
Lines 46-49 – While national-level data on salt intake is lacking, salt intake in the region is high and the study in northern Peru suggests a salt intake of 11g per day. It is better to highlight this to show why an intervention is required, although national-level data should be measured too.
Line 57 – The audience of interest hasn’t been appropriately defined or justified – why parents of young children?
Lines 77-78 – No justification given for involving high/low socioeconomic status districts, or private/public kindergartens
Lines 81-85 – this section is better suited to the background, as it helps set the context for the study
Line 94 – This sentence appears in the ethics section and so is not required here
Line 101 – How were the tools validated through discussion? Normal validation process involves testing in a representative sample of your study population?
Lines 135-136 – Justification for the number of focus groups should be provided in the methodology
Line 138 – why were men only involved in San Miguel?
Line 141 – Express the returned questionnaires as a participation rate
Line 147 – delete first part of sentence, unnecessary
Line 149 – does ‘studies’ mean qualification?
Line 150 – Sociodemographic status is mentioned throughout, but did you measure this according to income threshold, or simply employment/education level?
Line 158 – Half of participants?
Line 192 – ‘with 65.5% of men and 86.7% of women stating’ reads better than ‘since 65.5% of men and 86.7% of women indicated’
Line 193-194 – ‘as stated by 13.8% of men and 5.3% of women’ reads better than ‘who represented 13.8% according to the report of men and 5.3% by women’
Line 276 – insert ‘of’
Line 280 – insert ‘to’
Line 301 – ‘The methods participants were willing to use to lower’ instead of ‘What people were willing to do in order to adopt the desired behavior of lowering’
Table S1
should month read mother? ‘You would be willing to reduce salt intake in their food if it doesn’t sacrifice taste’Lines 372-377 – This section is more suited to the introduction as it provides context and justification for your study
Line 379 – insert ‘the’
Line 381 – Replace ‘In this sense’ with ‘Therefore’

Author Response
Please, find attached the document.

Reviewer 2 Report
Dear Authors,
It was a pleasure to read the paper, an interesting topic and very important. Please find some comments and suggestions.
Line 24 - '...results (84%), and female participants expressed that cooking was “their duty” as housewives.' please change to: results with 84% of female participants expressing that cooking was “their duty” as housewives.
Line 25-27 - 'Also during the qualitative phase, most considered their salt intake as adequate, and more than half mentioned that they have tried to reduce salt intake, the change in the flavor of the food being the most difficult challenge to continue with such practice.' please change to: The qualitative phase also revealed that despite the majority of the participants considered their salt intake as adequate a half of them mentioned that they have tried to reduce salt consumption, and the change in the flavor of the food was stated as the most difficult challenge to continue with such practice.
Line 30 - 'There were no salient differences in terms of responses..' do you mean no statistically significant differences between groups? if yes please say it
Line 37 - '...low and middle-income countries, ..' this is still a big issue in developed countries as well
Line 39 'In 2013, a meta-analysis reported that a reduction in salt intake lowers blood pressure and, thereby, reduces cardiovascular risk' yes, we no this, try to reword the sentence so it will sound more as being scientific-based - for example studies showed that 10% reduction of salt at the population level can reduce hypertension prevalence by ...
Lines 37 - 45 - please re-word these 2 paragraphs - they need to be linked and you are repeating the information
Line 46 - you mentioned that the information about salt intake in Peru is limited and you said '...level, e.g. one study in the north of Peru reported salt consumption levels of 11g/day (4.4g of sodium). is this the only study to report salt intake in Peru? if yes say this and do not use 'e.g.' if there is more data please mention the rage of salt intake reported in different studies
Line 78 - it is not clear why private and public nurseries were selected
Line 84 '...whilst transitioning to an omnivore diet.' - not sure what you want to say there; do you want to say that children were on the selective diet for some part of their life?
Lines 81 - 86 - this part should be incorporated into the introduction as a rationale for this study
Line 101 - please remove 'by three researchers of this study'
Line 103 - please remove ' For the first two, the following topics were covered' and replace with: The interview and focus group guides covered the following topics:...
Line 106 - remove ' illness'
Line 113 - remove 'who make-decisions about preparation of meals' replace by 'meals preparation and planning'
Line 125-126 - 'Data that was found to be different according to gender, district or type of school was stated accordingly.' I am not sure what do you want to say; statistically significant differences?
Line 126 - what statistical software has been used?
Line 141 - why did you expect to receive 400 questionnaires?
Line 147 please remove 'Considering the study phases, both qualitative and quantitative'
Table 2 - remove 'higher' before university - there are no higher and lower universities
Table 2 - what is 'Technical studies'? is it college?
Table 2 - 'Artificial seasonings' this term needs to be defined; do these seasonings contain sodium?
Line 164 - you stated 'in seven of the focus group discussions with women/ what about men? were there excluded from these discussions? if yes why?
Lines 169-171 - 'In the few cases which reported either being solely or jointly responsible for cooking or where it was a shared task, it appeared that the task of cooking went along with the decision of what was going to be cooked and buying the groceries' - this sentence needs to be rewritten - it is not clear what you want to say
lines 176 - 177 - you stated 'There were no clear differences in the responses provided by participants from different districts and different socio-economic groups.' and next you discuss some differences so, please reword this sentence' something lie that the majority of responses were similar however ...
Line 177 - remove 'would'
Line 222 - remove ' seasoning '
Line 231 - change 'know' to 'understand'
Line 233 - remove 'however, '
Line 268 ' artificial condiments' do they contain salt/sodium? do they use them to replace salt?
Line 320 - '21.6%' - this is not 'most'; what about the other 78% of participants what did they say about their salt intake?
Line 324 remove 'of' and replace with 'than'
Line 333 - remove 'a third' the responses shoed that 79.4% participants now the relation between salt and hypertension so it is more than 2 thirds
Line 377 - '...approximately 400 to 500 deaths from coronary heart disease over a decade of public health policies' - in the whole of England? over 10 years? if yes it was not the massive impact so I am not sure if it is a good example of efficient public health strategy
Lines 432 -435 - 'parents for in depth interviews to better capture their behaviors and insights related to decisions regarding food. This decision was made a priori on the assumption that fathers would not be available. This was later confirmed when conducting the focus groups, when there were fewer male participants than' - not sure what you would lie to say - there were male participants involved
Line 445 - 450 please remove this paragraph it is not needed in the conclusion
Line 454 - '... and will not require to segment for those of different socioeconomic status' to support this statement you will need to analyse the questionnaire by SES rather than by gender or both
Author Response
Please, find attached the document.

This manuscript is a resubmission of an earlier submission. The following is a list of the peer review reports and author responses from that submission.
Round 1
Reviewer 1 Report
It has been a pleasure to review this article. There are interesting findings here but you needed to succinctly and clearly describe the rationale for focusing on parents of preschoolers in this study. Your work needs extensive revision and sometimes it is difficult to follow. In particular, in the discussion section you should critically evaluate your findings in comparison to other previous work, and not only talk about the implications.
I have the following suggestions:
Abstract
It is too long. In Nutrients, the abstract should have about 200 words. I recommend abbreviating the abstract. I think that in the title or at least in the objective should go the study population that is being used because it is a very specific population: parents of pre-school children.
I think it should be introduced that you are going to study knowledge about salt and health.
Introduction
Line 46: specify which countries.
Line 49: a space between a number and a unit should be consistent through the manuscript.
Line 54 -55: please, could you specify the organizations that you talk about and which is the evidence that they present. It would be better to specify.
Line 63: please I don't understand this phrase, could it be reformulated?
Line 66: I think that you don't introduce why to study parents. You talk about individuals and the importance to study a community level but you are studying a specific population.
Methods
I think the methods part is the weakest part of the study or the one that is understood the worst. The specific cohort to be studied has not been previously introduced and we do not know the reasons for the study of parents instead of adults in general. I think the study population should be explained in more detail. It would be appropriate to see how many subjects are offered to participate in focus groups and interviews. Perhaps in order to make everything clearer, a figure could be made with the entry and retreat of subjects throughout the study. It is not very clear to me if women are always analyzed, or sometimes women and men (in the self-completed questionnaire). It could also be included in this figure which variables were studied or which was the objective to study in each of the stages of the study and mark the different stages.
Line 114: I do not understand how the questionnaire was validated.
Line 146: correct the double space.
Table 2: I think it is enough that you put the percentage and not the n of each subcategory.
I think it should be said that the interviews were conducted in Spanish (the language) but that the transcript of the recorded audios has been translated into English to be incorporated into the article.
Lines 187-195: I do not understand very well the percentage in relation to the n. Line 190, n=222 to which corresponds.
Line 231: slat = salt?
Line 249 - 254 : I don't understand the percentages, why 21% is related to 224 individuals (line 250) and (line 252), 31% is related to 215 individuals? maybe it could be incorporated a table to present the quantitative data.
Line 260 and 262: gaining weight is repeated.
Supplementary materials: you did not write the title of the supplementary material.
Reference: there are some mistakes, please you should review this references ( for example number 16 and 22.
Reviewer 2 Report
Dear Authors,
Thank you for submitting the manuscript entitled "Perceived barriers to reduce salt consumption and their implications to design a social marketing strategy in Peru: a mixed-methods study." The excessive consumption of salt is one of important public health problems across the world. Please find some comments and suggestion on how to improve the quality of the manuscript.
Title - the title does not really reflect the content of the anuscript; you do not necessary discuss barriers and their implications on a social marketing strategy. This manuscript is rather a general overview of the attitudes towards salt intake, which have a potential to inform future interventions.
"were no salient differences in terms" - I am to sure what this means?
Introduction:
"...the preparation of meals" - yes true but major contributor to salt intake is processed food accounting to 75% of total intake; is it different in Peru?
Methods
Study population - why this particular population has been chosen? why only parents of of preschool-aged children? this rather limit the study; as the sample may not be so representative
"The questionnaire was anonymous and their informed consent for it was not requested." - usually all studies with human participants including the use of questionnaires need signed consent; is it different in Peru?
Line 118 - rather health outcomes,
Results
Table 2 - column 'missing' what does it refer to? missing information about gender?
Please add a table to show quantitative data, also I recommend to use Chi2 test to analyse differences between men and women regarding the factors related to salt intake
I also recommend to analyse the focus groups data by gender (there were almost equal numbers of men and women) currently the results seems to looking at women responses only.
Line 250 - 21% considered their salt intake as low what about another 79%??
"On the other hand, the barriers to reduce their salt intake were the influence of family members and changes in food taste, but this response was supported only by one person" - I rather confused with this statement if it was a response of one person only it cannot be considered as barrier what about the responses of other people?
"...21% (n=224) of questionnaire participants endorse this statement." - I do not agree with this statement 21% is just a fifth of the group; what about another 79%?
"Self-perception about salt consumption" - this should not really be mentioned as a barrier as we do not know what is a current intake to compare the perception to; how could we know that self-perception is not correct? also have you evaluated willingness to change the salt intake by self-perception?
"Another barrier..." - why do you say another barrier? I cannot see any other barrier mentioned?
Discussion
Line 312 - "... but only a third reported knowing that there is an relationship between high salt consumption and hypertension." but in the results section you reported "The self-administered questionnaire found that 68% (n=231) of participants said hypertension was the most common illness due to a high salt intake..." so it rather more than one third?
Line 333 - "...natural seasonings and herbs were recognized by most of the participants, however, participants only mentioned a few number..." but in results section you mentioned "...56% (n=215) said using natural seasonings to cook."
Line 350 - "...prevent 400 to 500 deaths from coronary heart disease.." daily? yearly? in a specific country? worldwide?
Reviewer 3 Report
An interesting paper but it needs substantial amendments, as in its current form it does not provide adequate detail and information. The rationale for the study design and reason behind it needs further explanation. The paper also needs justification of the regions that were chosen, as it is currently not representative of the Peruvian population. Also needs the rationale for choosing private and public kindergartens.
Abstract doesn't really address the story, and the title somewhat misleads the reader. This is a knowledge, attitudes and behaviours type study, more so than a barriers one. It aims to assess knowledge, attitudes, and behaviours of parents with children of preschool age regarding salt and health.
The introduction needs to be stronger and could do with more background to salt - association with health and risk to cardiovascular disease among others. Current global recommendations and average population intakes in Peru (if there arent any, then state this and why). Any attempts made in Peru to address the salt issue - government initiatives for industry or awareness raising?
I have attached the manuscript pdf, with comments throughout the document for consideration.
